# Novel Submerged Photocatalytic Membrane Reactor for Treatment of Olive Mill Wastewaters

**Maria C. Fraga [1,2], Rosa M. Huertas [1,2], João G. Crespo [2] and Vanessa J. Pereira [1,3,*]**

[1]   iBET, Instituto de Biologia Experimental e Tecnológica, Apartado 12, 2781-901 Oeiras, Portugal; carmo.fraga@ibet.pt (M.C.F.); rosa.huertas@ibet.pt (R.M.H.)

[2]   REQUIMTE/Chemistry Department, Faculdade de Ciências e Tecnologia, Universidade NOVA de Lisboa, 2829-516 Caparica, Portugal; jgc@fct.unl.pt

[3]   Instituto de Tecnologia Química e Biológica António Xavier, Universidade Nova de Lisboa, Av. da República, 2780-157 Oeiras, Portugal

*   Correspondence: vanessap@ibet.pt; Tel.: +351-214469554 or +351-214469555

**Abstract:** A new hybrid photocatalytic membrane reactor that can easily be scaled-up was designed, assembled and used to test photocatalytic membranes developed using the sol–gel technique. Extremely high removals of total suspended solids, chemical oxygen demand, total organic carbon, phenolic and volatile compounds were obtained when the hybrid photocatalytic membrane reactor was used to treat olive mill wastewaters. The submerged photocatalytic membrane reactor proposed and the modified membranes represent a step forward towards the development of new advanced treatment technology able to cope with several water and wastewater contaminants.

**Keywords:** membrane filtration; ultraviolet radiation; photocatalytic membranes; hybrid reactor; olive mill wastewater treatment

## 1. Introduction

The treatment of wastewaters generated by the olive oil industry is a challenge, mainly in Mediterranean Countries. Their high content in solids, organic matter and phenolic compounds, make these wastewaters difficult to be treated by traditional methods. Processes such as membrane filtration [1] and $TiO_2$ photocatalysis [2] are alternatives that are being studied to treat these effluents.

High-quality permeates can be obtained when membrane processes are used to treat olive mill wastewaters [3,4]. However, the treatment of a highly concentrated retentate produced during membrane filtration and the development of fouling on the membrane surface are problems that still need to be addressed [5].

Several approaches can be employed to minimize the occurrence of fouling, such as an adequate pretreatment of the wastewater to be treated [6], optimization of chemical cleaning [7], the employment of backpulses or backwashes [3] and modification of the membrane surface in order to give superhydrophilic properties to the membrane [8]. Nanoparticles are widely used for this purpose, titanium dioxide ($TiO_2$) being the most studied due to its particular advantages [9]. The antifouling properties of $TiO_2$ are based on its strongly hydrophilic character [10] and ability to catalyze the degradation of organic substances [11], both enhanced in the presence of UV radiation.

$TiO_2$ photocatalysis can be performed in photocatalytic reactors with the photocatalyst suspended or immobilized on a carrier. Several schemes of photocatalytic membrane reactors are described and extensively reviewed elsewhere [12,13]. Regarding the membrane location, it can be placed outside in an external loop [14,15] or inside [16] the photocatalytic reactor. In the latter case, the system is defined as a submerged photocatalytic membrane reactor. In the literature, most of the work published

regarding the removal of organic compounds from water using submerged photocatalytic membrane reactors was performed with the photocatalyst in suspension [16–18].

When $TiO_2$ is in suspension, loss of $TiO_2$ due to adsorption to the system is expected [19]. Moreover, a further step is required in order to separate it from the treated water [12]. Furthermore, $TiO_2$ in suspension was reported to contribute to the development of fouling on the membrane surface [20–22]. In this context, the immobilization of $TiO_2$ on the membrane surface is the best solution for combining the two processes: photocatalysis and membrane separation. The immobilization of $TiO_2$ on the membrane surface was shown to be useful in the mitigation of fouling [23–27] and is expected to contribute to the degradation of compounds retained in the concentrated retentate produced.

The sol–gel process consists of a chemical process (hydrolysis-condensation) of a metal (or semi-metal) alkoxide precursor with itself creating a three-dimensional continuous solid linkage, through a basic or acid catalysis process [28]. This process has been proposed to synthetize $TiO_2$-based photocatalysts with high oxidation efficiency, as well as for $TiO_2$ immobilization in a large number of supports to control their porosity [29,30].

In a previous work, an extremely promising and reproducible photocatalytic activity was reported when silicon carbide surfaces were modified with $TiO_2$ obtained by the sol–gel process, using Degussa P25 and silicon carbide nanoparticles [31].

This work aims to prove that these membranes can be used to treat real olive mill wastewaters. Honeycomb commercial silicon carbide membranes were previously tested using the same matrix and proven to be extremely efficient in total suspended solids and oil and grease removal [3], ensuring compliance with the limits defined in the European legislation [32] for these two parameters. However, a higher removal of the dissolved fraction of chemical oxygen demand, total organic carbon and phenolic compounds is required to ensure that these parameters comply with current and future more stringent legislation. The modified photocatalytic membrane was tested in a new submerged photocatalytic membrane reactor conceived and assembled for this purpose, which can be easily scaled up. The efficiency of the individual and combined treatment processes was assessed.

## 2. Results and Discussion

### 2.1. Characterization of the Wastewater Matrix

The composition of the olive mill wastewaters is dependent on several factors such as type of extraction, type and degree of maturation of the olives, region and clima conditions [33].

Before each experiment, the pre-treated olive mill wastewater collected after sedimentation was characterized and Figure 1 depicts the range and average values obtained for each parameter assessed.

The volatile compounds detailed in Table 1 were identified in the wastewater matrix by comparing the mass spectra obtained for the different chromatogram peaks detected in each sample with mass spectra libraries (NIST 21, 27, 107, 147 and Wiley 229).

The peaks reported present the lowest similarity (between 84% and 98% similarity) between the library compounds and the mass spectra obtained. More compounds were identified in the chromatograms, but only the ones with percent area higher than 1% were considered.

Most of the compounds identified in this study have been previously reported in olives and olive oil by several authors [34,35], being responsible for the characteristic aroma and flavor of olive fruits and olive oil.

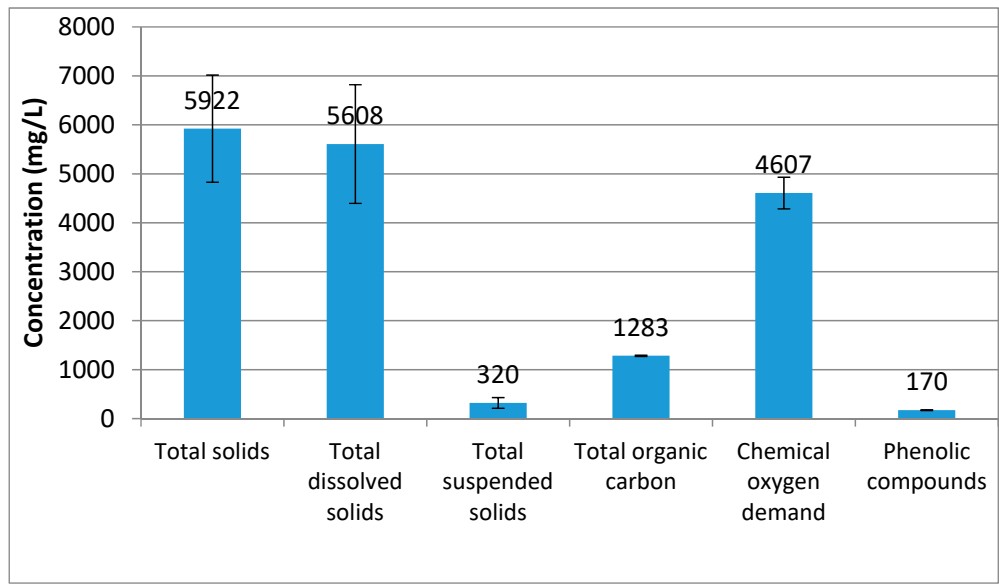

**Figure 1.** Characterization of the olive mill wastewater in terms of total solids, total dissolved solids, total suspended solids, total organic carbon, chemical oxygen demand and phenolic compounds; error bars present variation of wastewater characteristics measured in different sampling events.

**Table 1.** Volatile compounds detected in the feed samples.

| Compound | % Similarity * |
|---|---|
| n-buthyl-eter | 97 |
| 2-Hexanone | 95 |
| 2-octanone | 87 |
| 2-heptanone | 95 |
| trans-3,4-Epoxyoctane | 83 |
| Acetic acid | 93 |
| 2-nonanone | 94 |
| 2-buthoxyethanol | 96 |
| Propionic acid | 94 |
| Butyric Acid | 87 |
| Pentanoic acid | 93 |
| Hexanoic Acid | 96 |
| Heptanoic Acid | 94 |
| Cyclohexanecarboxylic acid | 89 |

* % similarity between the spectra and the libraries (NIST 21, 27, 107, 147 and Wiley 229); the values presented correspond to the lower values detected in the samples analyzed.

### 2.2. Determination of the Hydraulic Permeability and the Optimal Transmembrane Pressure Conditions

The hydraulic permeability was determined for the unmodified (control) membrane ($550 \pm 50$ Lh$^{-1}$m$^{-2}$bar$^{-1}$) and the modified membrane ($254 \pm 55$ Lh$^{-1}$m$^{-2}$bar$^{-1}$). Thus, the pore size reduction that the modified membrane presents led to a decrease of the hydraulic permeability. The intrinsic resistances of the membranes were calculated using Equation (1).

$$R_m = \frac{1}{Lp \times \mu} \tag{1}$$

where $R_m$ refers to the intrinsic resistance of the membrane, $L_p$ to the hydraulic permeability of the membrane and $\mu$ to the viscosity of water at the temperature of filtration. Values of resistance of $0.6 \times 10^9$ m$^{-1}$ to the control membrane and $1.2 \times 10^9$ m$^{-1}$ to the modified membrane were obtained.

In order to determine the optimum transmembrane pressure (the pressure gradient across the membrane that, in this system, is the pressure difference between the vessel and the filtrate) to use

in the filtration assays, different controlled transmembrane pressures were set for 5 min and the corresponding permeate flux values obtained with the real wastewater matrix recorded. Figure 2 shows the flux increase with the transmembrane pressure variation.

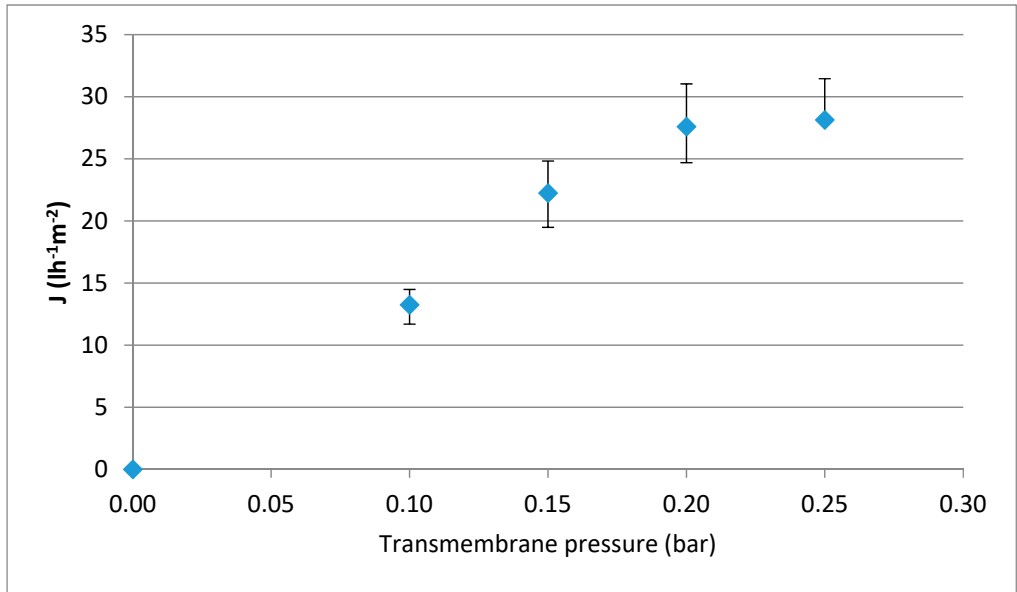

**Figure 2.** Determination of the optimal transmembrane pressure.

A value of transmembrane pressure of 0.2 bar was chosen since it was the highest transmembrane pressure value assuring the linearity between the transmembrane pressure and the permeate flux.

### 2.3. Wastewater Treatment in the Photocatalytic Membrane Reactor

Six filtration tests of 10 L of wastewater were performed during 4 h in the photocatalytic membrane reactor, using the unmodified membrane (control) and the modified photocatalytic membrane (Table 2). For each membrane, a test with the membrane in the presence of UV radiation was performed in order to evaluate the effect of direct photolysis and the photocatalytic activity of the membrane material; another test of membrane filtration in the absence of UV light was conducted in order to evaluate the rejection of compounds due to size exclusion and adsorption. This control was performed since the modification of the membrane leads to a pore size reduction and the different material of the top layer may affect the membrane adsorption.

Finally, a test combining filtration and UV radiation with the two membranes was carried out to evaluate the maximum potential of the hybrid combination. For all the tests, samples were taken each 20 min in the first hour and then each hour until the end of the test (4 h of assay). The hydraulic permeability of the control and modified membranes were measured before the experiments. After the filtration experiments, the membranes were cleaned with water at $60 \pm 5$ °C until at least 90% of their hydraulic permeability was restored. All the tests with the control and the photocatalytic membranes were performed with the same membrane to allow a reliable comparison between the different assays performed. In a previous study [31], the chemical stability (absence of titanium release) of the modified membranes was verified after filtration with different cleaning solutions at $65 \pm 5$ °C: distilled water, citric acid 2% (w/v) and NaOH 4% (w/v) whereas the reproducibility of the membranes was confirmed after five different degradation tests of methylene blue using two different commercial batches.

**Table 2.** Description of the six tests performed in the submerged photocatalytic membrane reactor.

| Test ID | Membrane Type | UV Light | Filtration | Objective |
|---------|---------------|----------|------------|-----------|
| C.UV | | Yes | No | Evaluate direct photolysis using low pressure UV lamps |
| C.F | Control | No | Yes | Evaluate the filtration performance of the control membrane |
| C.UVF | | Yes | Yes | Evaluate the combined effect of the control membrane |
| M.UV | | Yes | No | Evaluate the photocatalytic properties of the modified membrane surface |
| M.F | Modified | No | Yes | Evaluate the filtration performance of the modified membrane |
| M.UVF | | Yes | Yes | Evaluate the combined effect of the modified membrane |

In all the filtration tests, a permeability decrease was observed throughout the entire experimental time (Figure 3). An improvement of the membrane permeability was observed when the modified membrane was used in the presence of UV light, compared to the other filtration experiments conducted, mainly after 2 h of experiment (Figure 3). During the initial period of filtration, membrane fouling may occur at the membrane surface and also intrapore, due to penetration/adsorption of soluble material at the pores' inner walls. In the absence of a photocatalytic effect we expect that both outer surface and intrapore fouling play a role and permeability decline is severe. When a photocatalytic effect is present, outer surface fouling is expected to be reduced but intrapore fouling will probably not be minimized significantly. This explains why fouling is present in both situations, although with a better performance of the membrane where a photocatalytic activity is assured. This increase in permeability could be explained by the photo-induced antifouling or superhydrophilic properties of $TiO_2$ [10,36,37]. In the different experiments conducted (detailed in Table 2), the same batch of 60 L of real wastewater collected at a single sample event was used. However, since the real olive mill wastewater matrix used in this study is complex, the 10 L used in each assay are expected to vary in composition (as can be seen in the error bars of Figure 1 for the parameters monitored in this study). Small changes in composition may also have an effect in the membrane permeability profile so the changes observed cannot only be attributed to the different treatment processes under evaluation.

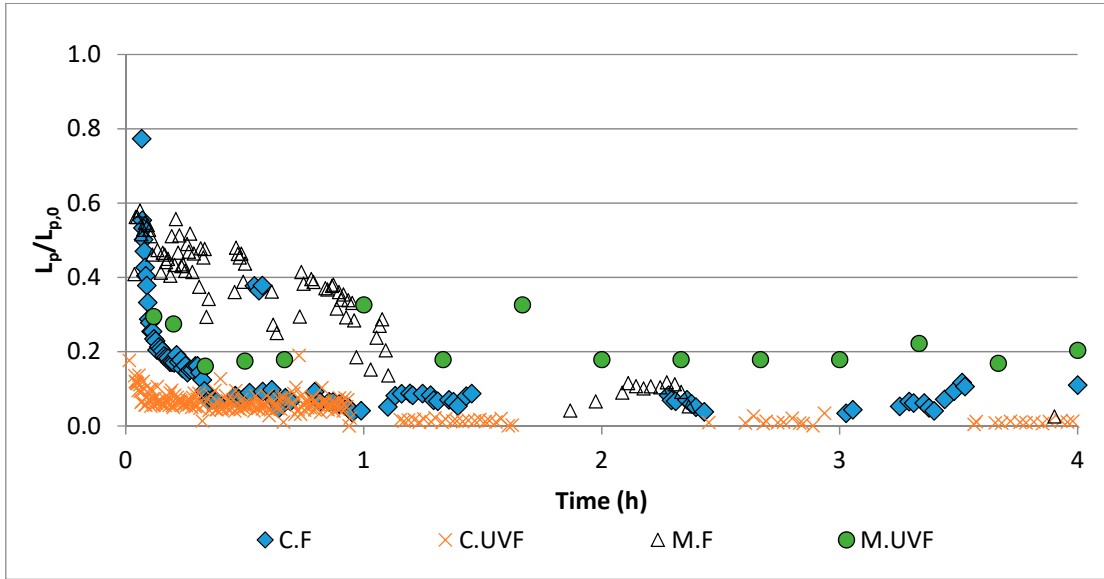

**Figure 3.** Variation of the normalized permeability of the membranes in the four filtration tests.

The total membrane resistances obtained in the end of the different assays, after 4 h of operation, were calculated using Equation (2):

$$R_t \ = \ R_m \ + \ R_f \ = \ \frac{TMP}{\mu_t \ x \ J} \tag{2}$$

where total membrane resistance ($R_t$), corresponds to the sum of the intrinsic membrane resistance ($R_m$) and the resistance due to fouling ($R_f$), *TMP* refers to the transmembrane pressure, *J* to the flux and $\mu_t$ to the fluid viscosity at working temperature (20 °C).

An analysis of the resistances due to fouling was performed in order to do a direct comparison between the two different membranes (control and modified membranes), taking into account their different intrinsic resistances ($R_m = 0.6 \times 10^9$ m$^{-1}$ for the control membrane; $R_m = 1.2 \times 10^9$ m$^{-1}$ for the modified membrane). Results obtained show a decrease in the resistance due to fouling when the modified membrane ($R_{\text{fouling, final}} = 5.8 \times 10^9$ at the end of the assay and average $R_{\text{fouling, 2–4 h}} = 6.5 \times 10^9$ of the last two hours) was used in the presence of UV radiation (test M.UVF) when compared with the control membrane ($R_{\text{fouling, final}} = 1.33 \times 10^{10}$ at the end of the assay and average $R_{\text{fouling, 2–4 h}} = 1.57 \times 10^{10}$ of the last two hours) in absence of UV radiation (test C.F), revealing that the hybrid system may have an effective impact in the mitigation of fouling since the advanced oxidation process contributes to the degradation of foulants present in the feed/retentate. It is important to note that a single assay was conducted using each test condition and that even though the same batch of real wastewater was used in all the experiments, the real olive mill wastewater matrix used in this study is complex and therefore the volume used in each assay is expected to vary in composition, which will also affect the resistance due to fouling.

Different parameters typically used to assess the water quality (total solids, total suspended solids, chemical oxygen demand, total organic carbon and phenolic compounds) were quantified in the different assays.

As expected, in the absence of filtration, direct photolysis did not affect the permeate solids composition (tests C.UV and M.UV, detailed in Table 2).

Total solids, total suspended solids and total dissolved solids were only quantified at the end of the assay (after 4 h of filtration) due to volume constraints of the samples. Table 3 shows the results obtained for the removal of these parameters.

**Table 3.** Removals of total solids, total suspended solids and total dissolved solids after 4 h of filtration.

| Test | % Removals after 4 h of Filtration Test | | |
|:---:|:---:|:---:|:---:|
| | **Total Solids** | **Total Suspended Solids** | **Total Dissolved Solids** |
| C.F | 14 | 93 | 8 |
| C.UVF | 14 | 92 | 10 |
| M.F | 17 | 89 | 13 |
| M.UVF | 20 | 98 | 18 |

The increase in the rejections obtained for the combined system using the modified membrane, compared with the combined system using the unmodified membrane, is probably due to the pore size reduction in the modified membrane and its photocatalytic activity.

The results of the removals obtained over time for chemical oxygen demand (COD), total organic carbon (TOC) and phenolic compounds in each filtration test (C.F, C.UVF, M.F, M.UVF) are depicted in Figure 4.

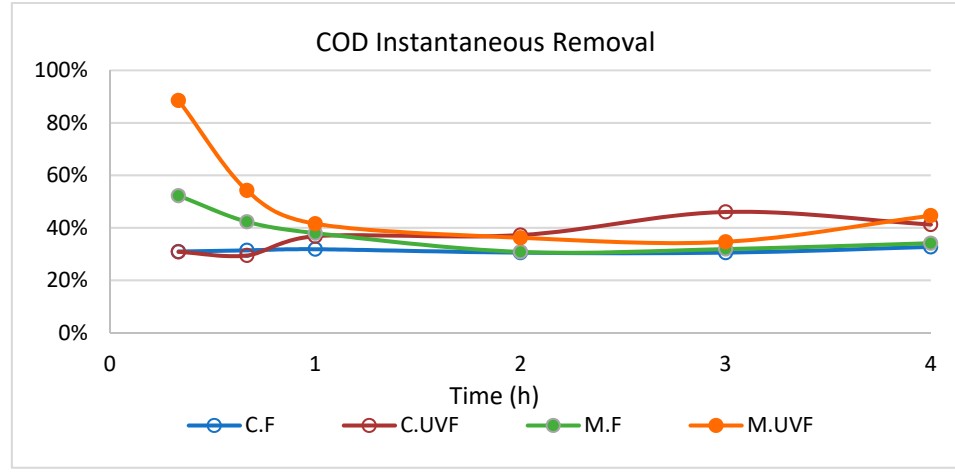

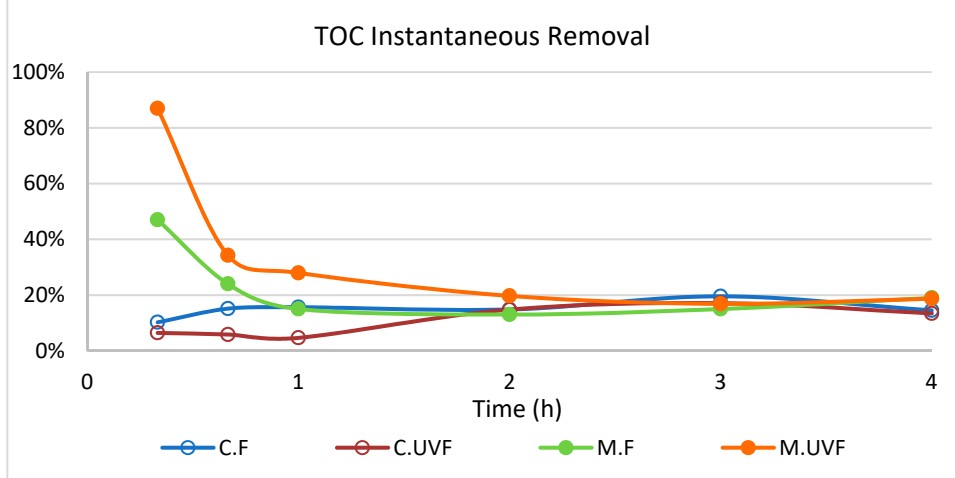

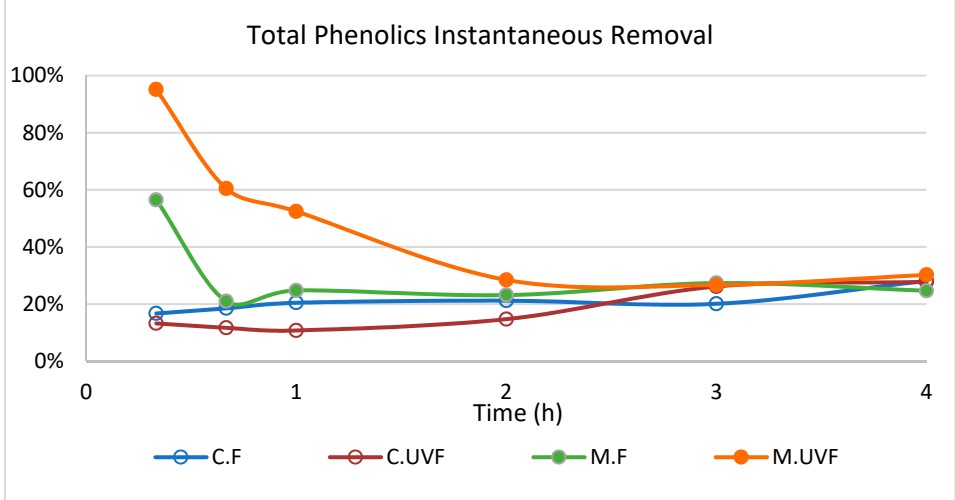

**Figure 4.** Percent removals of chemical oxygen demand (COD), total organic carbon (TOC) and phenolic compounds in the different tests: C.F, C.UVF, M.F and M.UVF.

When the control membrane was used in the absence of UV radiation (test C.F), removals with a constant trend were obtained, between 10% and 20% of total organic carbon, 31% and 37% of chemical oxygen demand and between 20% and 25% of total phenolic compounds. When the control membrane was used in the presence of UV radiation (test C.UVF), removals between 6% and 13% of total organic

carbon and 31% and 46% of chemical oxygen demand were obtained and between 13% and 29% of total phenolic compounds.

Results obtained in the test M.F showed that a reduction of the pore size of the membrane was achieved due to the surface modification. Removals of total organic carbon (47%), chemical oxygen demand (42%) and phenolic compounds (56%) were achieved in the first sampling time of the experiment, corresponding to 20 min of filtration, being considerable higher than the removals obtained in the same filtration time in the tests C.F and C.UVF. Nevertheless, a gradual decrease of the percent removals over time was observed until 1 h, and was maintained constant after this time, probably due to the accumulation of dissolved compounds on the membrane surface leading to their breakthrough through the membrane.

The filtration tests with the modified membrane conducted in the presence of UV radiation (test M.UVF) proved that the membrane has a significant photocatalytic activity since extremely high removals of total organic carbon (87%), chemical oxygen demand (89%) and phenolic compounds (95%) were achieved in the first sampling time of the experiment, corresponding to 20 min of reaction and filtration (Figure 5). In this case, the wastewater was forced to be in contact with the photocatalytic membrane surface due to the transmembrane pressure. Thus, the degradation of dissolved compounds was obtained due to the formation of free radicals—a consequence of the activation of the $TiO_2$ by the UV radiation. Therefore, a significant decrease of the analyzed compounds in the permeate was observed.

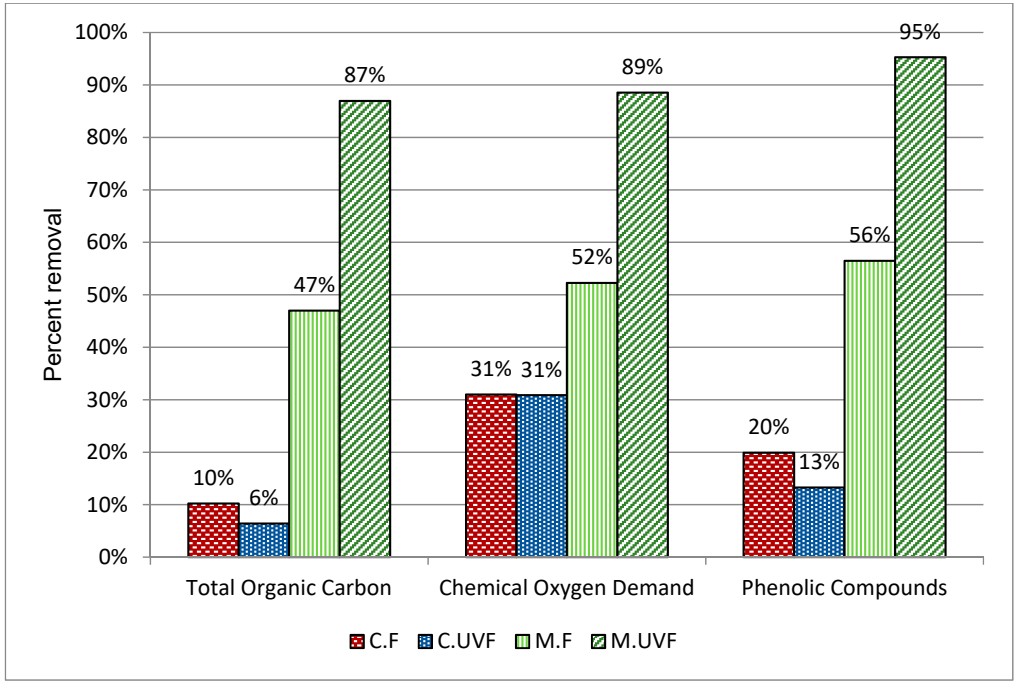

**Figure 5.** Percent removals of the analyzed compounds after 20 min of experiment.

These high removals gradually decreased until 2 h of experiment, and were maintained constant after this time. This fact could be explained by the building of a fouling layer and eventually a cake on the membrane surface, due to the high concentration of particulates in this specific effluent, which prevented the light to activate the photocatalytic layer during the whole period of the experiments. A consequent reduction of the photocatalytic activity of the membrane was observed, with a reduction of the degradation of these compounds, which was clear during the initial phase of these experiments. This problem could be overcome in a system built with an effective pre-treatment step or/and backpulse and/or backwash devices. These systems consist in very short pulses of air in the reverse direction to the permeation in the case of the backpulses, and pumping a given volume of permeate in the case of

backwashes. These systems were already tested in a previous work conducted in a pilot scale unit using unmodified honeycomb silicon carbide membranes to treat the same matrix and proved to be efficient in fouling mitigation [3], being thus a promising solution to overcome this problem.

The volatile compounds detailed in Table 4 were identified in the different assays by comparing a mass spectra library with the mass spectra obtained for the different chromatogram peaks detected in each initial feed and permeate collected after 20 min of filtration. The peaks reported present the highest similarity (between 84% and 98% similarity) between the library compounds and the mass spectra obtained.

A comparison between the obtained peak areas of the feed samples and permeates at 20 min of test was done in order to calculate the removals of each compound in the different tests. In the case of negative removals values, it can be considered that these compounds were formed during the tests as consequence of reactions non-monitored in this work, probably as a consequence of the aeration and direct photolysis. Since the objective of this research is to obtain treated water, removals as high as possible are desirable. Results show that n-buthyl-ether, trans-3,4-epoxyoctane and 2-nonanone were totally removed in all the tests, due to adsorption/rejection by both membranes, since their molecular weight is lower than the molecular weight cut off of the membranes.

An effect of degradation due to UV light was observed in the case of the 2-hexanone and 2-octanone since the removals increased in the presence of radiation with the control membrane.

The effect of the pore size reduction also led to an increase in the removals of these two compounds, observed when the modified membrane was used in the absence of radiation.

In the case of 2-hexanone, the effect of the photocatalytic membrane was observed, increasing the removals from 68% to 92%. A production of 2-heptatone in the presence of UV radiation was perceived. When the photocatalytic membrane was tested with UV radiation, 86% of acetic acid was removed, while in the other tests a production of this compound was observed. Regarding the hexanoic acid, heptanoic acid, cyclohexanecarboxylic acid and 2-buthoxyethanol, the use of the modified membrane enabled removals while production was observed with the control, being even higher in the presence of UV light.

In general, the modified membrane in the presence of UV radiation allowed achieving much higher removals of the identified volatile compounds (higher than 79% in all the cases, except 2-heptanone) with no production of any volatile compound. In fact, the degradation of organic volatile compounds by means of $TiO_2$ photocatalysis is widely studied and reported [38,39].

**Table 4.** Percent of similarity and removal of the organic volatile compounds detected in the feed and permeate samples collected after 20 min.

| Compound | C.F | | C.UVF | | M.F | | M.UVF | |
|---|---|---|---|---|---|---|---|---|
| | Similarity (%) | Removal (%) | Similarity (%) | Removal (%) | Similarity (%) | Removal (%) | Similarity (%) | Removal (%) |
| n-buthyl-eter | 97 | 100 | 97 | 100 | 96 | 100 | 97 | 100 |
| 2-Hexanone | 96−98 | 35 | 95−98 | 62 | 98 | 68 | 98 | 92 |
| 2-octanone | 87−91 | 37 | 87−95 | 100 | 98 | 98 | 87 | 100 |
| 2-heptanone | 95 | 100 | 91−96 | −104 | 98 | 79 | 97−98 | 44 |
| trans-3,4-epoxyoctane | 94 | 100 | 83 | 100 | 83 | 100 | 84 | 100 |
| Acetic acid | 93−98 | −25 | 98 | −25 | 95−98 | −72 | 94−98 | 86 |
| 2-nonanone | 94 | 100 | 97 | 100 | 98 | 100 | 98 | 100 |
| 2-buthoxyethanol | 96−98 | −27 | 96−97 | −46 | 95−98 | 67 | 98 | 100 |
| Propionic acid | 94−95 | −11 | 95−98 | −6 | 94−96 | 18 | 94−98 | 79 |
| Butiric Acid | 89−94 | 100 | 94 | 100 | 87−95 | 30 | 90−97 | 88 |
| Pentanoic acid | 93−96 | −8 | 94 | −2 | na | na | 96−98 | 91 |
| Hexanoic Acid | 94−96 | 6 | 96 | −22 | 96−98 | 85 | 96 | 94 |
| Heptanoic Acid | 98 | −57 | 94−96 | −123 | 96 | 61 | 94 | 100 |
| Cyclohexanecarboxylic acid | 84−89 | 2 | 83−84 | −3 | 89−90 | 42 | 89−90 | 89 |

## 3. Material and Methods

### 3.1. Submerged Photocatalytic Membrane Reactor

In this work, a new submerged photocatalytic membrane reactor (Figure 6) was developed and tested. The reactor is made of polyethylene. It has a cuboid shape with squared base with 19 cm width and length as well as a height of 30 cm. The membrane is placed in the center of the reactor. Two diaphragm pumps (12 V 3.0 A, 5.5bar; SZY-4155, Shui Zhi Yuan) ensure the pressure difference needed to achieve the intended permeate flux. A pressure sensor and controller (Aplisens, PCE 28, Warszawa, Poland) was used in order to measure and guarantee a constant pressure difference. The permeate flux was monitored using a pluviometer which indicates when 5 mL of permeate are collected. The data was continuously acquired by the software TeraTerm. The mixing of the system was done by an aeration of 0.33 L of air per liter of liquid per minute. The activation of the photocatalytic layer of the membrane is done through two low pressure UV lamps (Puro TAP, UVC, 11 W, type GPH212T5L, Christchurch, New Zealand) placed at 1.6 cm from each side of the membrane. The submerged flat sheet membranes (LiqTech International, Ballerup, Denmark) filter from the outside to the inside. The modified membranes can therefore be easily irradiated which will decrease fouling and increase the feed/retentate treatment due to direct and indirect photolysis. A hybrid module with multiple filtration elements and light sources can be easily assembled and therefore this proposed system can be easily scaled up.

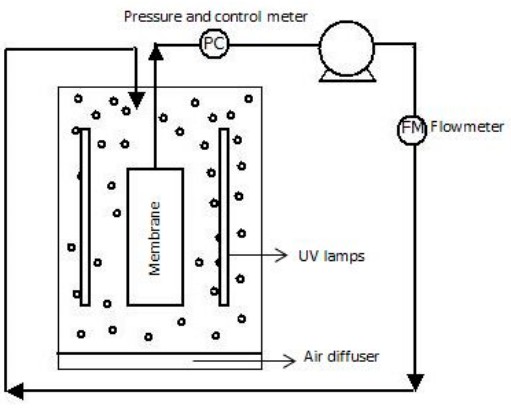

(**a**)

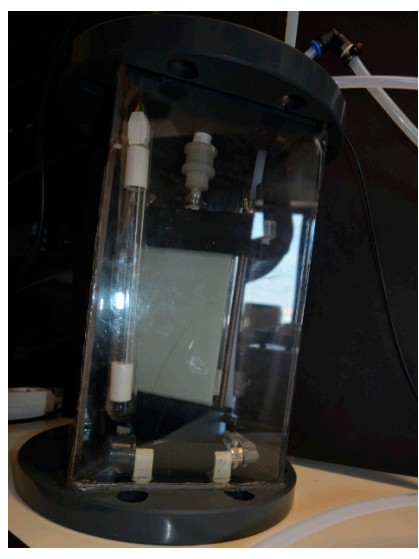

(**b**)

**Figure 6.** *Cont.*

Feed

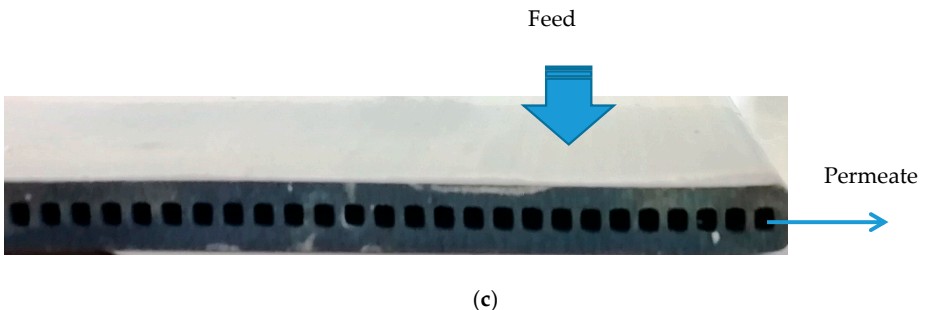

Permeate

(**c**)

**Figure 6.** Submerged membrane photocatalytic reactor (**a**,**b**) and photocatalytic membrane (**c**), developed and tested to treat olive mill wastewaters.

### 3.2. Wastewater Matrix and Analytical Methods

This work was conducted using a real olive mill wastewater matrix, collected at a wastewater treatment utility after the sedimentation process. The samples were characterized in terms of several parameters commonly used to evaluate the water quality using standard methods [40]: total solids (Standard Method 2540B), total suspended solids (Standard Method 2540D), chemical oxygen demand (COD) (Standard Method 5220) and total organic carbon (TOC) (Standard Method 5310B). Total dissolved solids were quantified subtracting the value obtained for total suspended solids from the value of total solids. Phenolic compounds were also quantified by the Folin–Ciocalteu method [41]. The wastewater matrix characterization results are shown in Figure 1.

The presence of volatile compounds was determined in different samples by solid phase microextraction (SPME) followed by gas chromatography mass spectrometry (GC/MS). The SPME procedure was conducted with a divinylbenzene/carboxen/polydimethylsiloxane fiber; (d = 50/30 μm; needle size: 23 Ga) from Supelco (Bellefonte, PA, USA). The extraction temperature was set to 40 °C, the agitation speed to 250 rpm, the extraction time to 40 min and the desorption time to 3 min. A Shimadzu QP 2010 GC/MS (Kyoto, Japan) with a Sapiens—Wax MS (TeknoKroma, Barcelona, Spain), 60 m, 0.25 mm (d.i.), 0.25 μm analytical column was used to detect the volatile compounds after splitless injection. The column temperature was held at 40 °C for 5 min, increased to 170 °C at a 5 °C/min rate, then to 230 °C at a 30 °C/min rate held for 4 min and finally to 270 °C at a 30 °C/min rate held for 5 min. The ion source and interface temperatures were set at 245 °C. The volatile compounds detected on the chromatograms were identified using the mass spectra libraries NIST 21, 27, 107, 147 and Wiley 229.

### 3.3. Preparation of the Photocatalytic Membrane

Commercial high flux flat sheet silicon carbide membranes (17 cm × 10 cm × 0.6 cm) from LiqTech International were used as substrates. Tetraethyl orthosilicate (TEOS, Sigma-Aldrich, 98%, Saint Louis, MO, USA) was used as precursor reagent in sol–gel preparation. Commercially available Degussa P25 titanium dioxide nanoparticles with 30–90 nm of nominal diameter were also employed in this study. All solvents employed were of reagent-grade quality and used without further purification. The modification protocol followed (labelled as SGSi-D (L$_3$)) was previously optimized and detailed by Huertas et al. [31]. Briefly, the concentration of Degussa TiO$_2$ nanoparticles used was 0.05 M and the molar ratio between TiO$_2$ and SiO$_2$ was 1:0.9. The control and modified membrane were previously characterized [31]. Compared to the control, the modified membrane had a higher number of pores, a higher pore density (2.5 ± 0.2 μm$^{-2}$), a lower porosity of 0.9% and lower average Feret diameter (longest distance between two points along the selection boundary) of 91 ± 8 nm. The active surface area in each side of the membrane is 145 cm$^2$. The only difference in this work was the deposition method since both sides of an entire membrane were modified in this study. Dip-coating was therefore used instead of the drop casting deposition method. After the inner membrane channels were sealed with silicone, the membranes were dipped in the sol–gel suspension for 30 s at a speed of 150 mm s$^{-1}$, and taken out at the same speed followed by drying at room temperature during 30 min and heating at

80 °C overnight. The procedure was repeated three times. The average thickness of the photocatalytic layer measured in 6 different areas was 10 ± 1 µm.

Homogeneous deposition of titanium and silicon were visualized (Figure 7), using a Bruker Quantax energy dispersive X-ray spectrometer (Bruker, Billerica, MA, USA) coupled to a SEM HITACHI (S-2400 model) scanning electron microscopy, after the modified membranes were sputter coated with an Au/Pd thin film using a South Bay E5100 apparatus (San Clemente, CA, USA).

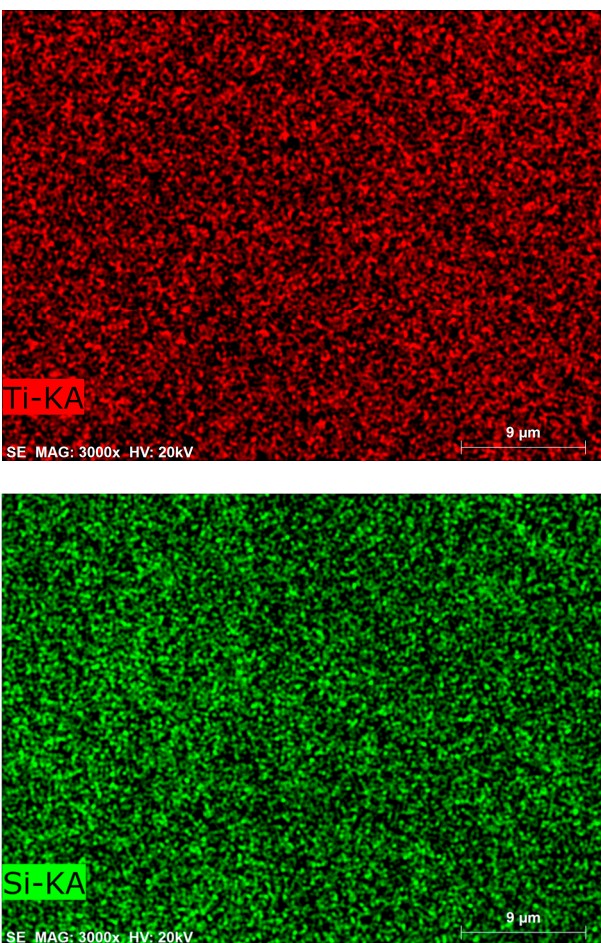

**Figure 7.** Mapping of titanium and silicon in the photocatalytic membranes prepared.

### 4. Conclusions

In this study, a new design of a photocatalytic membrane reactor was developed and tested to treat real olive mill wastewater and enhance the degradation of organic compounds. Results proved the photocatalytic activity of the membrane. Extremely high removals of chemical oxygen demand, total organic carbon and phenolic compounds were achieved with this system at 20 min of operation. Subsequent cake formation on the membrane surface, due to high concentration of particulates in this specific effluent, prevents the light from reaching the photocatalytic layer of the membrane with a consequent expected reduction of the permeate quality produced. It is suggested that this problem could be easily solved in a pilot/full scale system through effective pre-treatment and the application of strategies to minimize fouling, such as backwashing and backpulsing. These strategies were already proven effective and optimized when tubular unmodified silicon carbide membranes were tested in conventional pilot scale membrane filtration systems for the treatment of these wastewaters [3].

The proposed reactor can be easily scaled up and, with proper flux maintenance strategies to minimize cake build up, is expected to be an extremely efficient olive mill wastewater treatment process. The olive mill wastewater matrix is characterized by its high solids content and black color.

The proposed reactor is a promising approach to treat a multitude of drinking water and domestic wastewater matrices that are much less complex in terms of composition.

**Author Contributions:** Conceptualization, M.C.F., J.G.C. and V.J.P.; formal analysis, M.C.F.; investigation, M.C.F., R.M.H. and V.J.P.; methodology, M.C.F., R.M.H. and V.J.P.; supervision, J.G.C. and V.J.P.; validation, V.J.P.; Writing—Original Draft, M.C.F.; Writing—Review and Editing, J.G.C. and V.J.P.

**Funding:** The authors thank Adventech for supplying the wastewater matrices. iNOVA4Health—UID/Multi/04462/2013, a program financially supported by Fundação para a Ciência e Tecnologia/Ministério da Educação e Ciência, through national funds and co-funded by FEDER under the PT2020 Partnership Agreement is gratefully acknowledged. Associate Laboratory for Green Chemistry LAQV which is also financed by national funds from FCT/MEC (UID/QUI/50006/2013) and co-financed by the ERDF under the PT2020 Partnership Agreement (POCI-01-0145-FEDER—007265) is gratefully acknowledged.

**Conflicts of Interest:** The authors declare no conflict of interest.

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
