# Peer review of "Novel Submerged Photocatalytic Membrane Reactor for Treatment of Olive Mill Wastewaters"

_catalysts, doi:10.3390/catal9090769_

Round 1
Reviewer 1 Report
The manuscript focusses on the results obtained with a photocatalytic membrane reactor during treatment of olive mill wastewaters. The catalytic properties on the substrate membrane are induced by deposition of TiO2 particles. A series of results with and without TiO2 and with and without UV light irradiation are presented. This work is interesting, since it is based on the combination of two processes, UV degradation and membrane filtration, and their synergistic effects on the treatment of high-strength wastewaters. However, I think the paper has several weak points, which the authors need to improve before publication. Below I append more specific comments:
1. The graphical representation of results is not optimized. Especially in Fig. 3, it is not possible to distinguish the different evolutions. I recommend the authors to improve the presentation of data. In this case, I also wonder why the results with M.F are better than C.UVF. Also, why are the results with C.UVF that worse, as compared with C.F? I would have expected very similar results, since the control membrane does not have photocatalytic activity. Could the authors give an explanation for such results?
2. Which is the pressure shown in the x-axis of Fig. 2?
3. Which is the y-axis title in Fig. 5?
4. The sequence of each part, which might be a requisite of the journal, make difficult to understand some parts of the text, cause the experimental part is at the end of the manuscript.
5. Line 113: Authors wrote “An improvement of the membrane permeability during the filtration tests was observed when the modified membrane was used in the presence of UV light…” An improvement with respect to which conditions?
6. Authors analyze the effect of the experiment configuration on the reduction in resistance. Specifically, in line 129, they compare the resistance of the modified membrane with the control membrane. I found this comparison quite simplistic, considering that only one permeability value is given for the MF system at the end of the experiments. I also wonder if the authors did repetition of the experiments, or if they are just showing the results from single experiments. Would the results be reproducible and representative of the effects of the photocatalytic activity of the membranes?
7. Which is the loading of TiO2? Which is the active surface of the membrane?
8. Line 266, I guess it is Supelco.
9. Line 229. Is the reactor square or cylindrical? “The basis is a square with 19 cm diameter”
10. Which are the units vvm (line 235)?
11. In the end, I find that the results obtained with all membranes (Fig. 4) are almost identical, regardless of the type of configuration. Why are the authors just representing the results obtained after 20 minutes of filtration? Then, from line 182 one can read that the “building of a fouling layer prevented light to activate the photocatalytic layer”. From the introduction, it was supposed that the photocatalytic activity would be helpful to avoid the formation of a cake layer. Moreover, if a cake layer is formed with the M.UVF configuration, why is the resistance of this membrane not showing an increase in Fig. 3? These results seem somehow inconsistent. Also, once the membrane is fouled and cake layer is formed, would the photocatalytic activity disappear? Since only one experiment is shown, it is not clear if the photocatalytic activity is recoverable after conducting membrane cleaning. This aspect is very relevant for the applicability of the membrane reactor.
12. Line 23: is membrane filtration a new process?
13. Line 60: please change font type
Reviewer 2 Report
Reviews for “Novel submerged photocatalytic membrane reactor for treatment of olive mill wastewaters”
In this manuscript, the authors have modified a silicon carbide membrane by TiO2 via the sol-gel method to treat effluents in olive mill wastewaters by the photocatalytic reaction. The authors show that the TiO2 modified membrane effectively removes several chemicals in olive mill wastewaters after 20min UV light illumination. However, the manuscript will need a major improvement before published in Catalysts.
1. From Figure 4, as mentioned in the text, the TiO2 modified membrane only demonstrates a good performance within one hour of the test. The rest of 3 hours behaves very similarly to the controls. Although the authors provide the hypothesis of what could be the reason and the potential solutions to address the rapid dropping performance, it would not be enough to justify the initial experiment design. The authors merge the membrane and photocatalysts and try to take merits from the two components. The concept is excellent, but the overall results indicate that only limited chemicals can be decomposed, such as acetic acid and propionic acid. The membrane effective service time is modest. If the authors can perform the proposed back-pulse method, that can justify the whole experimental design.
2. Besides the permeability of the membranes, the authors don’t provide any characterization of the membrane, such as proximate pore size and surface area, etc.
Some other minor points
1. Line 59-60, “TiO2 obtained by …” should be the same font as the main text.
2. Several abbreviations are not explained, such as C.UVF, M.F, COD, TOC, etc.
3. Table 4 is at the end of the main text, while it has been mentioned before Table 2. Suggest reorganizing.
Round 2
Reviewer 1 Report
Authors have addressed all comments raised in my previous review. Although some weak points of the work are unavoidable due to the type of wastewater selected for the experiments, I think the paper can be accepted for publication.
Reviewer 2 Report
The author has fully addressed all previous comments in a very concise and professional way. The manuscript has been significantly improved and is ready to be published in Catalysts.